# Cholesterol, Amyloid Beta, Fructose, and LPS Influence ROS and ATP Concentrations and the Phagocytic Capacity of HMC3 Human Microglia Cell Line

**DOI:** 10.3390/ijms241210396

**Published:** 2023-06-20

**Authors:** Oscar M. Muñoz Herrera, Brian V. Hong, Ulises Ruiz Mendiola, Izumi Maezawa, Lee-Way Jin, Carlito B. Lebrilla, Danielle J. Harvey, Angela M. Zivkovic

**Affiliations:** 1Department of Nutrition, University of California, Davis, CA 95616, USA; ommunoz@ucdavis.edu (O.M.M.H.); bvhong@ucdavis.edu (B.V.H.); 2Department of Pathology and Laboratory Medicine, University of California, Davis Medical Center, Sacramento, CA 95817, USA; ruizmen@ucdavis.edu (U.R.M.); imaezawa@ucdavis.edu (I.M.); lwjin@ucdavis.edu (L.-W.J.); 3Department of Chemistry, University of California, Davis, CA 95616, USA; cblebrilla@ucdavis.edu; 4Department of Public Health Sciences, University of California, Davis, CA 95616, USA; djharvey@ucdavis.edu

**Keywords:** microglia, cholesterol, Alzheimer’s disease

## Abstract

Research has found that genes specific to microglia are among the strongest risk factors for Alzheimer’s disease (AD) and that microglia are critically involved in the etiology of AD. Thus, microglia are an important therapeutic target for novel approaches to the treatment of AD. High-throughput in vitro models to screen molecules for their effectiveness in reversing the pathogenic, pro-inflammatory microglia phenotype are needed. In this study, we used a multi-stimulant approach to test the usefulness of the human microglia cell 3 (HMC3) cell line, immortalized from a human fetal brain-derived primary microglia culture, in duplicating critical aspects of the dysfunctional microglia phenotype. HMC3 microglia were treated with cholesterol (Chol), amyloid beta oligomers (AβO), lipopolysaccharide (LPS), and fructose individually and in combination. HMC3 microglia demonstrated changes in morphology consistent with activation when treated with the combination of Chol + AβO + fructose + LPS. Multiple treatments increased the cellular content of Chol and cholesteryl esters (CE), but only the combination treatment of Chol + AβO + fructose + LPS increased mitochondrial Chol content. Microglia treated with combinations containing Chol + AβO had lower apolipoprotein E (ApoE) secretion, with the combination of Chol + AβO + fructose + LPS having the strongest effect. Combination treatment with Chol + AβO + fructose + LPS also induced APOE and TNF-α expression, reduced ATP production, increased reactive oxygen species (ROS) concentration, and reduced phagocytosis events. These findings suggest that HMC3 microglia treated with the combination of Chol + AβO + fructose + LPS may be a useful high-throughput screening model amenable to testing on 96-well plates to test potential therapeutics to improve microglial function in the context of AD.

## 1. Introduction

Microglia are known to be critically involved in the etiology of Alzheimer’s disease (AD) [1,2]. Homeostatic microglia are protective, participating in the clearance of amyloid beta (Aβ) and other cellular debris; on the other hand, activated or disease-associated microglia (DAM) drive neuroinflammation and neurodegeneration processes in the AD brain [3]. Due to this critical role of microglia in AD, and the failure of Aβ-based therapies to improve AD outcomes, microglia have emerged as an important therapeutic target to prevent and treat AD. Low cost, high-throughput screening in vitro models are needed to test a wide variety of molecules for their potential to reverse this deleterious DAM phenotype. It has been established that induced pluripotent stem cell (iPSC)-derived microglia are superior to cell lines and animal model-derived cells in replicating the complex, multi-faceted phenotypes of microglia in human brains and continue to be necessary to fully understand the mechanisms and underlying biology of microglia in the context of AD [4,5,6,7,8,9,10,11,12,13]. However, neither iPSC-derived microglia nor microglia isolated from animals are well-suited for high-throughput screening applications because they are expensive, difficult to grow and/or difficult to scale up. In this study, the objective was to determine whether the human microglia cell 3 (HMC3) cell line, which originates from immortalized human fetal brain-derived primary microglia culture, could act as a suitable model of DAM phenotype. Mimicking the DAM phenotype would generate a platform that could be used to screen large numbers of molecules for their ability to reverse specific dysfunctional aspects associated with activated microglia. A human cell line was selected due to known inter-species differences in critical aspects of microglia phenotype and function [5]. The HMC3 human microglia cell line was selected due to its documented ability to mimic the normal functions and responses of human microglia, including the expression and secretion of cytokines such as IL-6 both in the basal state and in response to stimulants such as Aβ, the production of reactive oxygen species (ROS), and ability to perform phagocytosis, among other properties [4]. Several different treatments alone and in combination with each other were assessed for their ability to generate critical aspects of the DAM phenotype, including alterations in apolipoprotein E (ApoE) expression and secretion, cytokine expression, mitochondrial dysfunction, and capacity to perform phagocytosis. The treatments included cholesterol (Chol), lipopolysaccharide (LPS), Aβ, and fructose. These treatments were chosen on the basis of the following observations.

Chol is critically involved in the pathophysiology of AD, from neurons, which are known to overproduce pathogenic Aβ peptides in the context of high plasma membrane Chol concentrations [14,15,16,17,18,19], to microglia, which are known to accumulate Chol and lipid-rich debris in multiple neurodegenerative conditions [20]. Genome-wide association studies point to Chol and microglia as key players in AD, with APOE, the main Chol transporter in the CNS, and TREM2, a monocyte-specific receptor, being among the strongest genetic risk factors for AD across populations [21]. When cells are faced with excess Chol, they can esterify it and store the resulting cholesteryl esters (CE) in lipid droplets. Lipid droplet accumulation is a hallmark of the AD brain [22]. The ability of microglia to remove Aβ is influenced by their cellular Chol clearance capacity [23]. Intracellular Aβ degradation is mediated by the Chol efflux function of ApoE [24]. DAM show reduced expression of homeostatic genes with an increase in lipid metabolism and phagocytosis genes [25]. In microglia, high Chol concentrations lead to induction of the DAM phenotype, in which increased inflammatory signaling, increased production of ROS, and decreased Chol efflux hinder the ability of microglia to clear Aβ, further increasing the concentration of Aβ oligomers and driving plaque formation [24,26].

Although the function of the blood–brain barrier (BBB) is to keep deleterious molecules from entering the brain, BBB function may be impaired in AD [27,28,29,30], and deleterious, pro-inflammatory molecules such as LPS have been found to cross the BBB [31,32]. High LPS concentrations induce cognitive impairment and neuroinflammation in the mouse brain [33]. In humans, a two-to-three-fold increase in LPS has been detected in the AD brain [34,35,36]. LPS is a potent pro-inflammatory activator of monocytic cells, including microglia.

AD is associated with multiple comorbidities, including diabetes and cardiovascular disease [37], with as many as 80% of AD patients developing diabetes [38]. Although hyperglycemia is, by definition, the primary metabolic dysregulation in diabetes, high fructose concentrations are a key driver of aberrant lipid accumulation in the liver [39,40], and high fructose intake has been found to be a causal factor in the development of insulin resistance and metabolic dyslipidemia [41]. Recently, short-term fructose intake was shown to impact hippocampal plasticity even in the absence of overt peripheral insulin resistance [42], and a new hypothesis linking brain fructose metabolism to the etiology of AD is emerging [43].

In this study, we address the hypothesis that HMC3 human microglia stimulated with multiple factors implicated in the pathogenesis of AD (i.e., Chol, LPS, Aβ, and fructose) replicate critical aspects of the DAM phenotype, including deficient cholesterol clearance, intracellular cholesterol accumulation, induction of a pro-inflammatory cytokine response, mitochondrial dysfunction, and reduced phagocytosis.

## 2. Results

### 2.1. Effects of Treatments on Expression of Pro-Inflammatory Cytokine Genes and APOE in HMC3 Microglia

Cytotoxicity experiments were first performed in order to determine appropriate doses and exposure time for Chol and Aβ treatment to minimize cell death (Appendix A). Appendix A demonstrate that a Chol treatment of 20 µg/mL for 6 h was a reasonable dose as it was the highest concentration that kept cell death below 10%. Aβ treatments alone demonstrated that Aβ was not significantly contributing to cell death at 24 h exposure; as a result, the 2 µM concentration was adopted from the literature [44].

HMC3 stimulation on 6-well plates is depicted by the diagram in Figure 1. As shown in Figure 2, treatments containing LPS generally resulted in an induction of tumor necrosis factor-alpha (TNF-α) and interleukin 6 (IL-6) expression. LPS and fructose + LPS also induced IL-1β; however, LPS alone and in combination with fructose, had no significant impact on ApoE expression. AβO-containing treatments and fructose alone, but not AβO alone, induced ApoE expression. AβO alone had no significant impact on gene expression within the scope of this analysis. Chol alone decreased ApoE expression, but in combination with other treatments expression of ApoE was generally increased. The combination treatment Chol + AβO + fructose + LPS induced ApoE and TNF-α expression the most; however, there was no significant effect with this treatment on IL-1β and IL-6.

### 2.2. Effects of Treatments on Whole-Cell Total and Esterified Cholesterol Concentrations, Mitochondrial Total Cholesterol Concentrations and ApoE Secretion in HMC3 Microglia

Whole-cell Chol content was increased by all AβO and Chol-containing treatments but not AβO or Chol alone (Figure 3A). The combination AβO + Chol increased whole-cell cholesterol significantly, and even more so with AβO + Chol + LPS and AβO + fructose + Chol + LPS (Figure 3A). Fructose alone, and in combination, increased cellular Chol the most (Figure 3A). AβO + fructose and fructose + LPS significantly increase whole-cell Chol concentrations, but treatment with fructose + Chol induced the highest concentrations (Figure 3A). LPS only contributed to an increase of whole-cell Chol concentrations when in combination with other treatments (Figure 3A). Esterified Chol concentrations were increased by AβO alone and fructose alone, as well as by all combination treatments, except AβO + Chol, LPS alone and Chol alone (Figure 3B). All instances of fructose treatments significantly increased esterified Chol in HMC3, but it was AβO + fructose + Chol + LPS that resulted in the strongest effect (Figure 3B). Mitochondrial total Chol content was significantly increased only by the combination of AβO + fructose + Chol + LPS (Figure 3C). In contrast to the observation of increased APOE gene expression (Figure 2), ApoE secretion was decreased in AβO-containing combination treatments, by the combination of fructose + Chol and especially by AβO alone (Figure 3D). AβO containing treatments resulted in the lowest concentrations of ApoE in the supernatant (Figure 3D). Chol alone, Fructose alone, LPS alone and fructose + LPS had no significant impact on ApoE secretion (Figure 3D).

### 2.3. Effects of Treatments on HMC3 Microglia ROS and ATP Concentrations and Phagocytic Activity

HMC3 stimulation on 96-well plates is depicted by the diagram in Figure 4. Although in most treatment conditions ATP concentrations were not significantly different compared to untreated cells, ATP production was significantly reduced by the combinations of AβO + Chol, AβO + Chol + LPS, and especially by the combination of AβO + fructose + Chol + LPS which reduced ATP production the most (Figure 5A). These observations suggest that, at these concentrations and exposure times, none of the treatments alone was capable of significantly altering ATP concentrations in the HMC3 cells. Only the combination treatments, all of which included AβO + Chol, significantly altered ATP concentrations. On the other hand, whole-cell ROS production was only significantly increased by AβO alone and by the combination of AβO + fructose + Chol + LPS, while all other treatments did not significantly modify whole-cell ROS (Figure 5B). Phagocytosis events, measured from the signal of internalized fluorescein-labeled *Escherichia coli* particles, were significantly lower in cells treated with AβO alone and AβO in combination with fructose and LPS (Figure 5C). AβO + fructose and AβO + LPS resulted in the lowest phagocytosis measurements (Figure 5C). LPS alone, fructose + LPS and AβO + fructose + Chol + LPS also reduced the fluorescent signal from internalized *Escherichia coli* particles (Figure 5C), whereas the effects of Chol, Fructose, AβO + Chol, Chol + LPS and fructose + Chol were not statistically different (Figure 5C).

### 2.4. Effects of Treatments on HMC3 Microglia Morphology

Untreated HMC3 microglia at time-point 0 h (Appendix A) can be described as elongated and spindly. HMC3 microglia following a 24-h period of treatment with the combination of AβO + fructose + Chol + LPS, on the other hand, can be described as contracted, semi-circular shapes (Appendix A). HMC3 microglia at the 24 h time-point, following AβO treatment, appear to roughly keep the elongated and spindly morphology (Appendix A). Similarly, HMC3 microglia at the 24 h time-point following Chol) AβO + Chol, fructose + Chol, and Chol + LPS treatment (Appendix A, respectively) also keep the elongated and spindly morphology. Following AβO + fructose, AβO + LPS, and AβO + Chol + LPS treatment (Appendix A, respectively), HMC3 microglia have a mixed morphology, with roughly 30% to 50% of the cell population abandoning the elongated and spindly morphology and adopting a contracted, semi-circular shape. Following fructose, LPS, and fructose + LPS treatment (Appendix A, respectively), HMC3 microglia have a contracted, semi-circular shape roughly 60% to 80% of the time, while the spindly and elongated morphology makes up the minority of the microglial population.

## 3. Discussion

AD is defined by the accumulation of Aβ [45], and DAM were defined by their colocalization with Aβ plaques [46]. Thus, Aβ is the most obvious mediator of the DAM phenotype. Aβ deposition induces the expression of neurotoxic pathways, including pro-inflammatory cytokines [47]. In our study, AβO alone reduced the concentration of ApoE secretion to one of the lowest concentrations and increased whole-cell esterified Chol concentrations, in addition to increasing ROS concentrations and reducing phagocytic activity. AβO-containing treatments (AβO + LPS, AβO + Chol, AβO + fructose, and AβO + Chol + LPS) induced APOE expression but reduced ApoE secretion, while also increasing whole-cell total Chol and CE. AβO + fructose and AβO + LPS additionally reduced phagocytosis.

However, there are additional factors that are known to drive pathophysiology in AD, and in fact, the amyloid hypothesis has recently come into question in the AD field due to emerging evidence that some individuals can have no neurological symptoms even in the presence of very high Aβ plaque burden [48], and due to recent failures and controversies with regard to the effectiveness of drugs targeting Aβ [49]. Microglia are critically involved in the pathophysiology of AD in their multiple roles, from the secretion of neurotrophic factors to clearance of Aβ, and have been found to have aberrant transcriptional and functional profiles in the brains of AD patients [50]. AD is likely a multi-factorial disease with poor management of concentrations of additional factors such as fructose, LPS, and Chol, which also play roles in the development of microglia dysfunction [37,38,39,40,41,42,43]. In fact, in our experiments, the biggest changes were induced by the combination treatment containing all four stimulants (AβO + fructose + Chol + LPS), which increased cytokine expression and increased APOE expression but resulted in one of the lowest levels of ApoE secretion and some of the highest concentrations of whole-cell total Chol, CE, and mitochondrial Chol concentrations, as well as increasing ROS concentrations to the highest level, reducing ATP concentrations to the lowest level, and reducing phagocytic activity.

APOE is highly expressed in microglia, and APOE4 is shown to promote the neurodegeneration-associated inflammatory phenotype of mouse microglia [51] and alter the functions of human microglia-like cells (iMGLs) [52]. It has been suggested that APOE4 confers a pro-inflammatory DAM phenotype [52,53]. ApoE4 carriers have lower cholesterol efflux capacity and lower mitochondrial function [54]. APOE4 iMGLs are fundamentally unable to mount normal microglial functionality when compared to APOE3 and APOE4 genotype-impaired phagocytosis and migration and aggravated inflammatory responses of iMGLs [55]. Future studies are needed to determine the APOE genotype of the HMC3 microglia cell line.

In previous work, it was shown that primary microglia from mice fed a diet constituting 29% fat, 34% sucrose, and 1.25% cholesterol (*w*/*w*) plus 42 g/L glucose and fructose (55%/45%, *w*/*w*) in drinking water had increased expression of IL-6 and TNF-α [56]. Microglia energy metabolism is an area of research that is emerging as critical in developing a better understanding of the pathogenesis of AD. It is now more and more appreciated that the metabolic reprogramming that occurs in microglia, depending on the available nutrients and environment, drives microglia phenotype and function [57]. Because microglia account for as much as 16% of brain cells, more so in plaque areas where microglia concentrations are increased, microglia contribute significantly to overall nutrient utilization, and the specific role of their metabolism in regulating neuroinflammation is of great interest [57]. Importantly, microglia are the only cell type in the brain that express the GLUT5 transporter, which is specific for the transport of fructose [58], suggesting that fructose may play an important as yet poorly defined role in microglia-mediated role in overall brain energetic metabolism. It is important to note that plasma fructose concentrations are increased in patients with type 2 diabetes [59] and that in multiple animal models, high fructose intake has been linked with an increased risk for dementia and decreased cognitive function [60]. In our study, fructose alone induced APOE expression, while Chol alone dampened APOE and IL-1β expression in microglia. Additionally, fructose alone increased whole-cell total Chol and CE content. The combination of Chol + fructose induced APOE expression; however, this combination reduced ApoE secretion to similar concentrations as AβO + fructose + Chol + LPS. Chol + fructose also increased whole-cell total and esterified Chol to even greater concentrations than their individual effects. However, it was only the AβO + fructose + Chol + LPS treatment that increased mitochondrial total Chol concentrations. ATP concentration was only significantly reduced by the combined treatment with Chol + fructose and not by either individually, and phagocytosis was only significantly reduced with AβO + fructose + Chol + LPS. When the expression of rate-limiting enzymes in glucose metabolism is inhibited in primary microglia and in the BV2 microglia cell line, fructose-6-phosphate inhibits ATP production and phagocytosis [61]. However, in this study we did not find a significant reduction in ATP concentration or phagocytosis when HMC3 were treated with fructose alone; however, the combination treatment including Chol + AbO + fructose + LPS did decrease both ATP production and phagocytosis events.

In our study, LPS-containing treatments induced the expression of pro-inflammatory cytokines. iMGLs robustly responded to LPS with significant induction in all measured cytokines (IL-6, TNFα, among others) [55], similarly observed in HMC3 microglia [62]. TNF-α, IL-6, and IL-1β are also expressed in primary human microglia when treated with LPS and when treated with Aβ 1–42 during a 6 h period [63]. However, in our experiments, we did not see the expression of these genes with AβO alone. LPS-treated HMC3 cells were previously shown to exhibit an appearance characteristic of activated microglia that generally had larger cell bodies, fewer branches, and an amoeboid phenotype [62]. In our experiments, AβO + LPS and AβO + Chol + LPS treatments shift a portion of the microglia to that amoeboid phenotype. An even greater percentage of microglia experience that same morphological shift with the LPS treatment, in agreement with Baek et al., and the fructose + LPS treatment. However, the potency of the morphological shift away from the elongated and spindly to the semi-circular, fried-egg morphology was better achieved with the combination treatment of Chol + AbO + fructose + LPS. LPS stimulation in the BV2 microglia cell line did not result in an increase of total and esterified Chol levels [64], which is in agreement with our observations in HMC3 microglia that LPS alone did not influence cellular Chol content.

Microglia have the ability to be dynamic in their response to an equally dynamic plethora of stimuli that happen in the brain [65]. The literature suggests that microglia morphology is indicative of the often complex information these monocytic cells are receiving from their surroundings [66]. The result appears to be a wide range of states the cell can adopt in order to process the information; they can be ramified, de-ramified, amoeboid-like, rod-like, and everything in between [66,67]. Future studies using a combination of instruments and software could help distinguish between these morphologies and, at the same time, identify where and how these morphologies overlap so that, ultimately, this library of data can help us better understand how microglia morphology is related to their functional profile in studies such as this one [68].

There are several limitations to this study. First, although our findings demonstrate the potential utility of this multi-stimulant approach using the HMC3 microglia cell line to mimic critical aspects of the DAM phenotype, additional aspects of the DAM phenotype, for example, the expression of cell surface markers such as CD68, were not measured. Additionally, the impact of molecules such as beta-catenin, arginase, and brain-derived neurotrophic factor should also be explored because of their relevance in blood–brain barrier health. Further validation and testing are needed to evaluate more thoroughly which aspects of the DAM phenotype are vs. are not replicated by this approach and, therefore, which target pathways would be suitable for molecular screening purposes using this approach. The results and observations described here highlight that a multi-stimulant approach may be useful to replicate the conditions present in the brains of AD patients, with cholesterol accumulation in conjunction with pro-inflammatory activation together leading to impairment in microglia function. However, further studies are needed to confirm the underlying mechanisms of these interactions in HMC3 cells and how well these replicate the known alterations in pathways involved in the development of the DAM phenotype in human patient samples [53]. For example, future studies should examine whether HMC3 microglia stimulated with Chol + AbO + fructose + LPS have increased expression of lipoprotein lipase and whether the observed changes in cytokine gene expression are also observed at the level of cytokine production [20].

We demonstrate in this paper that HMC3 microglia, a verified human cell line [4], treated with a combination of Chol + AbO + fructose + LPS, replicate multiple critical aspects of the DAM phenotype, including increased expression of pro-inflammatory cytokines, increased intracellular Chol accumulation, increased mitochondrial Chol accumulation, decreased ApoE secretion, decreased mitochondrial function, and decreased phagocytosis activity. Thus, HMC3 microglia treated with Chol + AbO + fructose + LPS is a promising tool that could be used in high-throughput screening platforms to test potential therapeutics compatible with this 24 h treatment window [69,70]. The advantage of this tool for high-throughput screening is several-fold: (1) the HMC3 microglia cell line is easy and affordable to grow and maintain, (2) treatment with the combination of Chol + AbO + fructose + LPS appears to adequately reproduce critical aspects of the DAM phenotype observed in AD, and (3) the readouts (total cellular Chol, ATP, and ROS content, ApoE content in the supernatant, phagocytic activity and cell morphology) are also easy and affordable to measure. Future studies can now test this new screening tool to discover candidate molecules capable of reversing the induced cellular dysfunction, which can then be further tested for their effectiveness with more sophisticated methodologies, including iPSC cells and animal models.

## 4. Materials and Methods

### 4.1. Study Design

Two sets of experiments were conducted using the HMC3 human microglia cell line purchased from the American Type Culture Collection (ATCC, Manassas, VA, USA, CRL-3304) to evaluate the effects of treatment with Aβ oligomer (AβO), fructose, Chol, and LPS, individually and in combination with each other. One set of experiments was performed on 6-well plates and the second set on 96-well plates. Both approaches, prior to analysis, differ only by the number of cells seeded per well, where microglia for the 6-well plates (GenClone, San Diego, CA, USA, 25–105 MP) were seeded at 400,000 cells per 2 mL, per well; and the microglia on 96-well microplates (Corning, Union City, CA, USA; Costar; 3916) were seeded at 40,000 cells per 200 µL, per well. In both sets of experiments, HMC3 microglia were cultured using EMEM (ATCC, Manassas, VA, USA, 30-2003), penicillin–streptomycin (10,000 U/mL, Thermo Fisher, Carlsbad, CA, USA, 15140122), and 10% fetal bovine serum (FBS, ATCC, Manassas, VA, USA, 30-2020). Seeded cells were allowed to adhere for at least 7 h. Following adherence, media was replaced with EMEM without phenol red (Thermo Scientific, Carlsbad, CA, USA, C837K00) to reduce background signal, penicillin–streptomycin (10,000 U/mL, Thermo Fisher, Carlsbad, CA, USA, 15140122) and 10% fetal bovine serum (FBS, ATCC, Manassas, VA, USA, 30-2020), along with the corresponding treatment, where applicable.

Microglia were treated with AβO (1–42) (HFIP-treated) (BACHEM, Vista, CA, USA, H-7442) at 2 µM for 24 h using a dosage and incubation time as previously described [44], or fructose (Millipore Sigma, Carlsbad, CA, USA, Fructose F0550000 100 MG) at 50 mM based on previously published dosage and incubation time [71] for 24 h, or the combination of both. At the 18 h time-point, the HMC3 were loaded with water-soluble Chol (Sigma–Aldrich, St. Louis, MO, USA, C4951) at 20 µg/mL, for 6 h. At the 21 h time-point, the cells were treated with LPS (Millipore Sigma, Carlsbad, CA, USA, L4005-100 MG) at 100 ng/mL for 3 h using a dosage and incubation time based on previously published literature [62]. At 24 h, the supernatant in all wells was removed and stored at −80 °C for analysis. Additionally, in all instances of Chol quantification, lysing was performed using a solvent prepared with chloroform (Sigma–Aldrich, St. Louis, MO, USA, C2432), isopropyl alcohol (Fisher Scientific, Carlsbad, CA, USA, 67-63-0) and Np40/Igepal Ca 630 (Sigma–Aldrich, St. Louis, MO, USA, 9002-93-1), at 7:11:0.1, respectively.

The first set of experiments performed with 2 technical and 2 biological replicates required the use of 6-well plates to generate sufficient material for the analysis of gene expression by qPCR, cellular and mitochondrial Chol concentrations, and ApoE content in the supernatant. Immediately following the treatments, the supernatant was removed and stored at −80 °C. The microglia were harvested for RNA extraction and qPCR analysis and for mitochondrial isolation. The supernatant was probed for ApoE content. Chol measurements were performed on lysed samples of whole-cell microglia and lysed isolated mitochondria.

For qPCR analysis, cells were harvested using a Cell scraper, 2-position blade, size: M (SARSTEDT, Newton, NC, USA, 83.3951). Total RNA from cultured cells was extracted using RNeasy Plus Mini Kit (QIAGEN, Redwood City, CA, USA, 74134). cDNA was synthesized using 100 ng RNA and iScript Reverse Transcription Supermix (BioRad, Hercules, CA, USA, 1708841). qPCR was performed using SsoFast™ EvaGreen Supermix and CFX96 qPCR system (BioRad, Hercules, CA, USA, 1725201 and 1845096). The forward/reverse primer sequences used are listed in Table 1. Gene expression was normalized to an endogenous gene, β-actin. Relative cDNA levels for the target genes were analyzed by the 2^−ΔΔCt^ method.

Mitochondria were isolated using a commercially available kit (with Dounce Homogenizer) (abcam, Eugene, OR, USA, ab110171) following the manufacturer’s instructions. The cellular content of unesterified and esterified Chol was measured using the Total Cholesterol Assay (Cell Biolabs, Inc., San Diego, CA, USA, STA-390) following the manufacturer’s instructions. A portion of the culture supernatant was probed for ApoE content using Human ApoE ELISA (Cell Biolabs, Inc., San Diego, CA, USA, STA-367) following the manufacturer’s instructions.

The second set of experiments was conducted on 96-well plates with 3 technical and 3 biological replicates, as recommended by the manufacturer, to measure the mitochondrial function and phagocytosis capacity. Immediately following the treatments, the supernatant was removed in order to perform ATP quantification (abcam, Eugene, OR, USA, ab83355), ROS measurement (abcam, Eugene, OR, USA, ab113851 DCFDA/H2DCFDA), and Phagocytosis activity (Molecular Probes, Eugene, OR, USA, Vybrant, V-6694) according to manufacturer’s instructions for each kit.

HMC3 cytotoxicity analyses were performed using Promega CellTox™ Green Cytotoxicity Assay (Madison, WI, USA, Cat#: G8743), using the End-point method on 96-well plates, at 40,000 cells per 200 µL per well (Appendix A). Chol, C4951, was tested in a range of 0, 5, 10, 20, 30, 40, and 50 µg/mL for 6 h, as well as 10 and 20 µg/mL for 24 h. AβO (1–42), H-7442, was tested in a range of 0.0, 0.1, 0.25, and 0.5 µM for 24 h.

To assess cellular morphology, pictures were taken using the Summit SK2-14X 14.0MP PC/MAC Digital Microscope Camera (OptixCam, Roanoke, VA, USA, TMS-SK2-14X).

Positive controls for DAM cellular states were not explicitly included in the study design given that an in vitro model of the DAM state has not yet been developed, and indeed, the specific mechanisms of how homeostatic microglia transform into DAM have not yet been clearly identified [72].

### 4.2. Statistical Analysis

Experiments using 6-well plates were performed twice (two biological replicates) in order to generate sufficient material for cellular RNA extraction and qPCR analysis and quantifiable amounts of cellular mitochondria when isolated. For experiments in the 96-well plates format, three biological replicates were performed since the manufacturer instructions for the ATP, ROS, and phagocytosis assays recommended triplicates in order to achieve the expected inter- and intra-plate variability of the kits.

Data were analyzed on R version 4.1.1 (R Foundation for Statistical Computing, Vienna, Austria). We inspected for normality using the Shapiro–Wilk test. Data that was non-normally distributed were log2 transformed. For qPCR analysis, the 2^−ΔΔCT^ method was used to analyze the relative changes in gene expression. For multiple comparison analyses, an ANOVA post-hoc Dunnett’s test was performed to compare treatments to the control. Results were considered statistically significant at *p* < 0.05. Statistical significance is between each treatment compared to the control. Bars represent SD.

All data are available in the Appendix A.

## Figures and Tables

**Figure 1 ijms-24-10396-f001:**
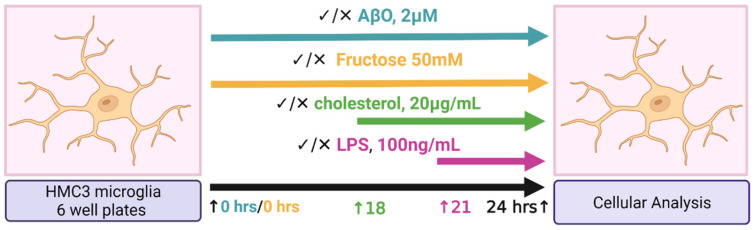
Diagram depicting HCM3 microglia treatments under a 6-well plate format. HMC3 microglia were treated with Aβ-oligomer (1–42) (AβO, 2 µM, 24 h, or fructose at 50 mM for 24 h, or the combination of both. At the 18 h time-point, the HMC3 were loaded with water-soluble cholesterol at 20 µg/mL, for 6 h. At the 21 h time-point, the cells were treated with lipopolysaccharide (LPS) at 100 ng/mL for 3 h. Image generated with BioRender.

**Figure 2 ijms-24-10396-f002:**
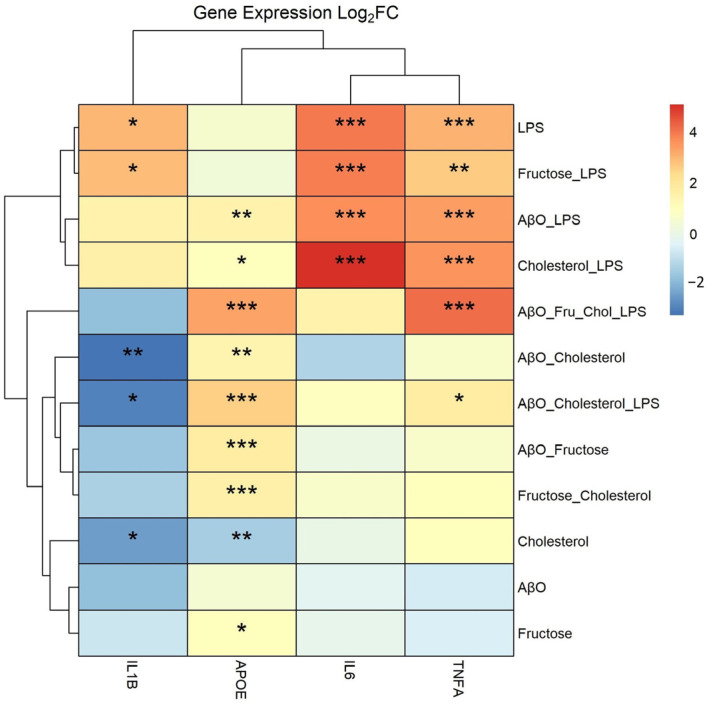
Effects of treatments on expression of pro-inflammatory cytokine genes and APOE in HMC3 microglia. Heat map from qPCR analysis from treated HMC3 microglia on 6-well plate. Microglia were treated with AβO at 2 µM for 24 h, or fructose at 50 mM for 24 h, or the combination of both. At the 18 h time-point, the HMC3 were loaded with water-soluble Chol at 20 µg/mL, for 6 h. At the 21 h time-point, the cells were treated with LPS at 100 ng/mL for 3 h. Terms: Lipopolysaccharide (LPS), amyloid beta oligomer (AβO), cholesterol (Chol), fructose (Fru), tumor necrosis factor-alpha (TNFA), interleukin 6 (IL6), interleukin 1β (IL1B), apolipoprotein E (APOE) expression. *N* = 2, two separate plates for two biological replicates. Data presented as fold change, * *p* < 0.05, ** *p* < 0.01, *** *p* < 0.001. Statistical significance is between each treatment compared to the control.

**Figure 3 ijms-24-10396-f003:**
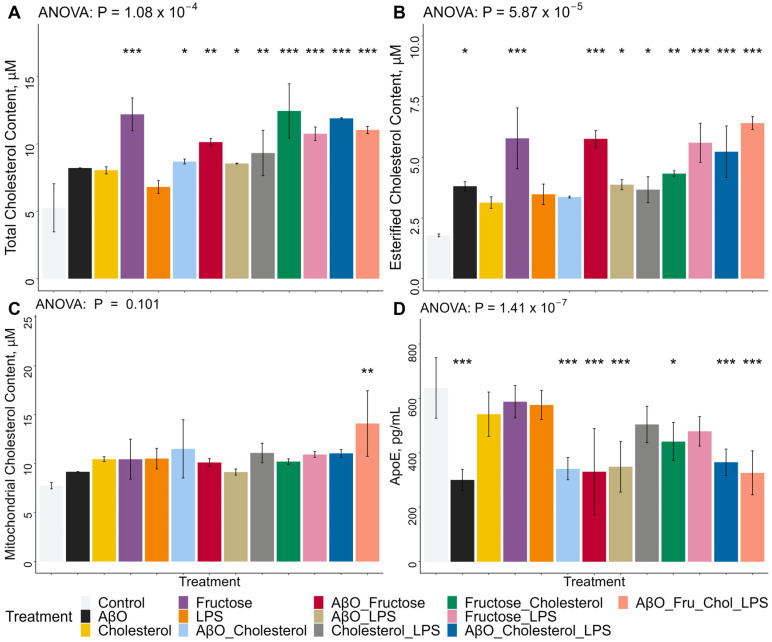
Effects of treatments on whole-cell total and esterified cholesterol concentrations, mitochondrial total cholesterol concentrations, and ApoE secretion in HMC3 microglia. Microglia were treated with AβO at 2 µM for 24 h, or fructose at 50 mM for 24 h, or the combination of both. At the 18 h time-point, the HMC3 were loaded with water-soluble Chol at 20 µg/mL, for 6 h. At the 21 h time-point, the cells were treated with LPS at 100 ng/mL for 3 h. (**A**) Total cholesterol concentration measurement performed on lysed whole-HMC3 microglia following treatments on 6-well plates. Fluorescence readings were compared to the cholesterol standard curve. *n* = 2, two separate plates for two biological replicates. (**B**) Esterified cholesterol concentration calculated from the fluorescent reading of the reaction without the esterase enzymatic component; performed on lysed whole-HMC3 microglia following treatments on 6-well plates. Fluorescence readings were compared to the cholesterol standard curve. *n* = 2, two separate plates for two biological replicates. (**C**) Mitochondria total cholesterol concentration measurement performed on lysed isolated HMC3 microglial-mitochondria following treatments on 6-well plates. Fluorescence readings were compared to the cholesterol standard curve. *n* = 2, two separate plates for two biological replicates. (**D**) APOE ELISA quantification performed on the supernatant from treated HMC3 microglia on 6-well plate. Fluorescence readings were compared to the human ApoE standard curve. *n* = 2, two separate plates for two biological replicates. Lipopolysaccharide (LPS), amyloid beta oligomer (AβO), cholesterol (Chol), fructose (Fru), apolipoprotein E (ApoE) protein. Data presented mean ± SD unpaired two-tailed *t* test, * *p* < 0.05, ** *p* < 0.01, *** *p* < 0.001. Statistical significance is between each treatment compared to the control. Bars represent SD.

**Figure 4 ijms-24-10396-f004:**
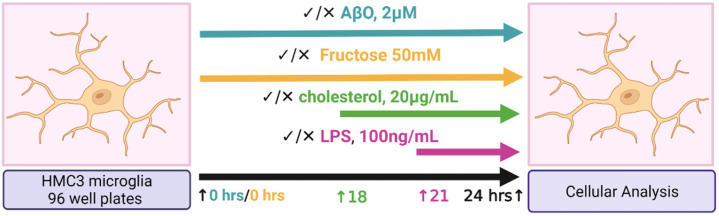
Diagram depicting HCM3 microglia treatments under a 96-well plate format. HMC3 microglia were treated with Aβ-oligomer (1–42) (AβO, 2 µM, 24 h, or fructose at 50 mM for 24 h, or the combination of both. At the 18 h time-point, the HMC3 were loaded with water soluble cholesterol at 20 µg/mL, for 6 h. At the 21 h time-point, the cells were treated with lipopolysaccharide (LPS) at 100 ng/mL for 3 h. Image generated with BioRender.

**Figure 5 ijms-24-10396-f005:**
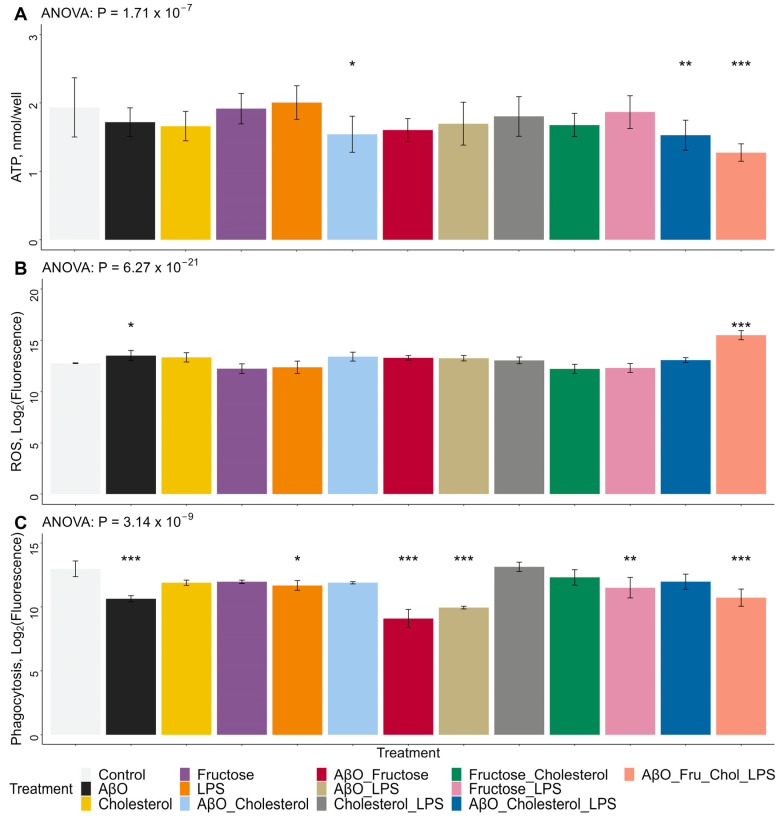
Effects of treatments on HMC3 microglia ROS and ATP concentrations and phagocytic activity. Microglia were treated with AβO at 2 µM for 24 h, or fructose at 50 mM for 24 h, or the combination of both. At the 18 h time-point, the HMC3 were loaded with water soluble Chol at 20 µg/mL, for 6 h. At the 21 h time-point, the cells were treated with LPS at 100 ng/mL for 3 h. (**A**) Whole-cell adenosine triphosphate (ATP) fluorescence measurements on 24-h period treated HMC3 microglia on 96-well plates. Fluorescence readings were compared to the ATP standard curve. *n* = 3, three separate plates for three biological replicates. (**B**) Whole-cell reactive oxygen species (ROS) fluorescence measurements by reaction with DCFDA/H2DCFDA, on 24-h period treated HMC3 microglia on 96-well plates. *n* = 3, three separate plates for three biological replicates. (**C**) Phagocytosis activity, quantified by fluorescein-labeled Escherichia coli (K-12 strain) readings on 24-h period treated HMC3 microglia on 96-well plates. *n* = 3, three separate plates for three biological replicates. Terms: Lipopolysaccharide (LPS), amyloid beta oligomer (AβO), cholesterol (Chol), fructose (Fru). Data presented mean ± SD unpaired two-* *p* < 0.05, ** *p* < 0.01, *** *p* < 0.001. Statistical significance is between each treatment compared to the control. Bars represent SD.

**Table 1 ijms-24-10396-t001:** Primers and their sequences used for qPCR analysis. Interleukin 1β (IL-1β), interleukin 6 (IL-6), tumor necrosis factor-alpha (TNF-α), apolipoprotein E (ApoE).

Gene	Primer Sequence (Invitrogen, Waltham, MA, USA)
IL-1β (Human)	FW: CCACAGACCTTCCAGGAGAATG RV: GTGCAGTTCAGTGATCGTACAGG
IL-6 (Human)	FW: CCAGCTATGAACTCCTTCTC RV: GCTTGTTCCTCACATCTCTC
TNF-α (Human)	FW: CTCTTCTGCCTGCTGCACTTTG RV: ATGGGCTACAGGCTTGTCACTC
β-Actin (Human)	FW: TCAAGATCATTGCTCCTCCTGAG RV: ACATCTGCTGGAAGGTGGACA
Gene	Primer Sequence (Bio-Rad, Hercules, CA, USA)
ApoE (Human)	qHsaCED0044297

## Data Availability

All of the data are available in a Appendix A.

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
