# Peer review of "Cholesterol, Amyloid Beta, Fructose, and LPS Influence ROS and ATP Concentrations and the Phagocytic Capacity of HMC3 Human Microglia Cell Line"

_ijms, 2023, doi:10.3390/ijms241210396_

Round 1

Reviewer 1 Report

            The manuscript submitted by Herrera et al. describes the response of the human microglia cell 3 (HMC3) cell line to various stimuli. The authors’ goal was to determine whether HMC3 microglia were suitable for high-throughput screening in the development of Alzheimer’s disease therapeutics. They find the combined treatment of HMC3 cells with cholesterol, oligomeric Aβ, fructose, and LPS best recapitulates the disease associated microglial (DAM) phenotype across several assays. The experiments are well-performed and appropriate. There are a few items that would strengthen the manuscript for publication. General and specific comments follow below.

General Comments:

·         The data presented is relevant to the field and the manuscript is well structured, starting with cytotoxicity assays for their stimulants, moving through studies of gene expression, cholesterol content, apoE secretion, ROS production, and phagocytosis, and ending with morphology.

·         Cited references are relevant, and more than half (66%) are recent (within the last 5 years). There is not an excessive number of self-citations.

·         The manuscript appears to be scientifically sound, with the experimental design appropriate to test the hypothesis.

·         The figures are appropriate. They are easily read, understood, and interpreted. The data is interpreted appropriately and consistently throughout the manuscript.

·         The conclusions are consistent with the evidence and arguments presented.

·         The statements regarding ethics and data availability are absent.

Specific Comments:

·         The authors do a nice job explaining the treatment paradigm, however, the inclusion of a schematic depicting the paradigm (Figure 1 panel A?) would make it easier for the reader to visualize and understand.

·         In the “2.2. Statistical Analysis” second of the Methods, there is no explanation for sample size/replicate numbers or power analysis. Please include justification for the experiments being performed twice (two biological replicates) using 6-well plates and three times (three biological replicates) using 96-well plates.

·         In section 3.3 “Effects of treatments on HMC3 microglia ROS and ATP concentrations and phagocytic activity” there is mention of “the combination of fructose+Chol+LPS (Figure 3C).” This combination is not included in any of the graphs, and if it is referring to the combination of Aβo+fructose+Chol+LPS, it is redundant with the following sentences. Please amend to accurately reflect the data being referenced and reduce confusion.

·         Section 3.4 examines the effect of treatment on MHC3 morphology. The authors describe the shape of the cells under various conditions and refer to Supplemental Figure 4B. While their description appears to be accurate, an inset and/or higher magnification images is needed to see this for certain.

·         Along similar lines as the above point, while morphology doesn’t correspond precisely to microglial activation state, it can provide useful information. Please expand the rationale and discussion of what these morphological changes mean. Furthermore, why not perform Sholl analysis to quantify the changes? This could elevate this data from supplemental to a main figure.

·         Finally, Supplemental Figure 5 could be better assembled for easier viewing (such as with Supplemental Figure 4), rather than individual images with corresponding descriptions below.  

Reviewer 2 Report

intro:

The introduction is really clear and provide enough information. 

The aims and hypothesis are both well described. 

Materials and methods section is flawless, the techniques are well described, the concentrations are chosen from previous published articles.

I would suggest to develop more the role of microglia in AD. This review can help: DOI: https://doi.org/10.1124/pharmrev.121.000400 

Results: 

The results are convincing, and the results are clearly presented and are not misleading. 

I would suggest to quantify some markers other than inflammatory molecule (example: BDNF, Beta Catenin, Arginase,...) and/or molecules involved in the BBB protection since the major outcome in AD the the BBB disruption. It would be interesting to evaluate the activation level of these cells. 

Discussion: 

You mention "the expression of cell surface markers such as CD68, were not meas- ured." I think it's really easy to evaluate in vitro using Westernblot. I would suggest to do it. 
